# TIEM: Enhancing Explanation of Video Prediction via Temporal Dynamics-Focused Dual Perturbation

## Abstract

Explaining video data predictions is challenging due to the complex spatio-temporal information in videos. In particular, the existing perturbation-based methods for video interpretation often fail to consider different temporal contexts, making them ineffective for dynamic videos where the important regions change rapidly or appear ephemerally across frames. To address this, we propose a novel video interpretation method, time importance score-aware extremal perturbation masks (TIEM), that enhances explainability by focusing on temporal dynamics in videos. TIEM exploits a dual perturbation process: first, it evaluates temporal importance across frames via temporal perturbation and then generates spatio-temporal extremal perturbation masks using the temporal importance explicitly. Our experimental results demonstrate that TIEM resolves the key challenges of the existing methods, providing more precise explanations across the time domain in synthetic white-box models and black-box models for real-world videos.

## 1 Introduction

Artificial intelligence (AI)-based predictive models have been widely used across various domains such as healthcare, finance, autonomous driving, and video analysis, especially due to the powerful predictive performance of deep neural network (DNN) models (LeCun et al., 2015; Yu et al., 2020; Ashfaq et al., 2022). However, despite their widespread use, these DNN models are often referred to as "black-box models," meaning that their internal workings are not transparent to humans. This lack of transparency makes it difficult to trust their predictions and therefore difficult to use them for decision-making as well (Guidotti et al., 2018; Buhrmester et al., 2021). To address this issue, a range of explainable AI (XAI) techniques are being developed that interpret and explain how the predictive models work internally (Adadi & Berrada, 2018; Samek & Müller, 2019). A visual explanation method, one of the representative XAI techniques, is used to explain black-box models for images and videos by visualizing the input regions that influence the model's output (Selvaraju et al., 2017; Chattopadhay et al., 2018; Alicioglu & Sun, 2022).

So far, research on visual explanation has focused primarily on interpreting the model's predictions for a single image, which leads to visually clear explanations of the predictions(Adebayo et al., 2018). However, visual explanation methods for a single image are difficult to directly apply to prediction models that use video data as its input, since the video data is not a simple collection of multiple images. The images that make up video data have complex recurrent spatio-temporal dependencies, such as the order and connections between the images and the interactions between objects within the images (Zhou et al., 2018). Therefore, to interpret the predictions for a video, visual explanation methods need to consider not only the spatial information in the images, but also the temporal information across the images.

To illustrate how temporal information in a video provides insight to humans, Fig. 1 shows the frames composing a "tennis swing" video. The images presented at the top of the figure visualize a video composed of the four frames arranged in the correct order of the original video, while the images at the bottom visualize a video composed of the same frames, but their order is shuffled. In the figure, the original video contains both spatial and temporal information across the four frames, allowing us to recognize it as a video of a "tennis swing." On the other hand, the shuffled video



Figure 1: Example of video to illustrate temporal information.

contains the same spatial information as the original video, but the temporal information is missing. As a result, with the shuffled video, it is difficult to tell whether the video shows the player swinging, serving, or simply waiting for the ball. This example clearly shows that the temporal information in a video, based on the sequence, interaction, and connection of frames in different temporal contexts, plays a crucial role in the predictions for videos.

## 1.1 MOTIVATION AND CONTRIBUTION

In the recent development of XAI methods for video predictions, there have been significant efforts to consider the temporal information inherent in video data. To this end, the existing perturbation-based works (Li et al., 2021; Uchiyama et al., 2023) mainly attempt to address the temporal information in videos. They extend the visual explanation method for a single image by expanding the dimensions for the time domain, but not considering it explicitly. As shown in the example in Fig. 1, the interactions of the spatial information between the consecutive frames are significant for the effective estimation of the temporal information. To consider this observation, the existing works estimate the temporal information by blending the spatial information of the adjacent multiple frames in the time domain. As a result, this blending-based approach can generate a natural and smooth visual explanation. In particular, it is effective in interpreting the predictions for *gentle* videos with low temporal dynamics, in which the important regions across frames change gradually in the spatial domain, e.g., a video in which important objects in the video shift gently across the frames (Li et al., 2021).

However, such a blending-based approach makes the estimation of the temporal information highly dependent on the spatial information across the adjacent frames. This may lead to an over-integration of the spatial and temporal information and a failure to consider temporal contexts of different lengths. As a result, the blending-based approach may be ineffective in interpreting the prediction for *dynamic* videos with high temporal dynamics, in which the important regions change rapidly or appear ephemerally across frames.

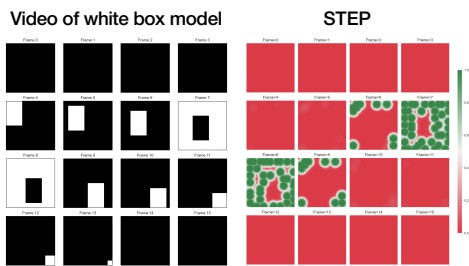

Figure 2: Example of a white-box model. The white regions are the ground truth of its visual explanation. The frame numbers start from 0 and increase from left to right, top to bottom.

Here, we classify the issues that may arise due to the limitation of the blending-based approach into *temporal concentration* and *temporal spillover*. To clearly describe the issues, we provide an example of a visual explanation for a synthetic white-box model by using STEP in Fig. 2. In the visual explanation, the important regions are highlighted in green. The white-box model generates random numbers for each pixel and computes predictions using only the numbers of the white regions. Therefore, its ground truth for visual explanation is the white region in each frame. In the example, the white-box model is designed so that the important regions change rapidly and appear ephemerally in frames 7 and 8, while the important regions gently shift across frames 4-6 and 9-13.

Temporal concentration indicates a phenomenon in which the estimated important regions are overly concentrated on specific frames that have an excessive influence on the prediction. This also leads to the frames near the excessively influential frames being overlooked in the interpretation. In the example, this issue is clearly shown as most of the estimated important regions are concentrated in frames 7 and 8, leading to the neglect of the white regions in frames 4-6 and 9-13. In addition, temporal spillover refers to a phenomenon in which the estimated important regions in one frame spill over into its adjacent frames. This issue is observed in frames 6 and 9 of the example. In those frames, the edges are estimated as important regions as in frames 7 and 8, even though they do not lie within the ground truth region. In real-world applications, these issues can occur in videos, where a particular action is present in the video for only a short time, such as sports and accident safety. The results and discussion of the issues in real-world datasets are presented in Section 4.2.

In this paper, we propose a novel video interpretation method based on perturbation, called time importance score-aware extremal perturbation masks (TIEM). To overcome the above issues, it considers the temporal dimension separately, through dual perturbation. More specifically, TIEM first analyzes how much each frame is significant for the prediction by using temporal perturbation, in terms of various window lengths. Then, it evaluates the time importance score (TIS) of each frame by combining the analyses of the frame for the window lengths. The TIS enables TIEM to estimate the temporal importance of each frame considering its temporal relations to the adjacent frames in different temporal contexts. Finally, TIEM finds a spatio-temporal visual explanation by conducting the TIS-aware spatial analysis for each frame based on extremal perturbation. This novel dual perturbation of TIEM not only avoids the over-integration of spatial and temporal information across the adjacent frames but also estimates the temporal dynamics more precisely. Through the experiments with the synthetic white-box model and the black-box model for real-world datasets, it is shown that the proposed method resolves both temporal concentration and spillover. In addition, it outperforms the state-of-the-art video interpretation method, especially in terms of temporal interpretation.

## 2 RELATED WORKS

Recent research on visual explanation methods has moved beyond interpreting a single image, and increasingly focused on incorporating temporal information to interpret the predictions for video data. The early works (Chattopadhay et al., 2018) interpret the predictions considering the multiple frames that make up a video, but do not consider the temporal information across frames. In the subsequent works, a variety of visual explanation methods have been developed that incorporate the temporal information during interpretation to achieve more accurate video interpretation and visualization as summarized in Table 1.

Table 1: Visual explanation methods for video prediction

| Study | Type | Interpretable model | Temporal awareness |
|-------|------|---------------------|--------------------|
| Bargal et al. (2018) | Gradient | CNN-RNN | Temporal normalization |
| Stergiou et al. (2019a) | Gradient | 3D-CNN | Feature pyramids |
| Stergiou et al. (2019b) | Gradient | 3D-CNN | Spatio-temporal saliency map |
| Hartley et al. (2022) | Gradient | 3D-CNN | Computing superpixel |
| Uchiyama et al. (2023) | Perturbation | 3D-CNN | Optical flow |
| Li et al. (2021) | Perturbation | Model-agnostic | Smoothing |
| Ours (TIEM) | Dual perturbation | Model-agnostic | Time importance score |

In Bargal et al. (2018), a gradient-based interpretation method, called cEB-R, is proposed that incorporates temporal information during video interpretation. It extends the excitation backpropagation method by introducing temporal normalization during backpropagation. Another gradient-based method, Saliency Tube (Stergiou et al., 2019b), leverages 3D saliency maps to consider the temporal information in 3D CNNs. In addition, Class Feature Pyramids (Stergiou et al., 2019a) consider different kernels at varying network depths within a 3D CNN model to reflect temporal information. SWAG-V (Hartley et al., 2022), which extends the SWAG method designed for interpreting a single image, integrates temporal information by creating superpixels from the model's gradient values. The superpixel in SWAG-V incorporates temporal characteristics.

As another approach, perturbation-based methods (Li et al., 2021; Uchiyama et al., 2023) have been proposed, which are similar to our work. In Li et al. (2021), a spatio-temporal extremal perturbation (STEP) method is developed, which is the only model-agnostic method for video prediction. It extends the extremal perturbation (EP) technique (Fong et al., 2019) by additionally considering the time domain, thereby considering temporal information during interpretation. Furthermore, it uses a 3D kernel to smooth and limit identified important regions across frames. In Uchiyama et al. (2023), an adaptive occlusion sensitivity analysis (AOSA) framework is proposed that reflects an optical flow between frames during its perturbation-based importance analysis. This enables the AOSA framework to incorporate temporal information, rather than simply extending the existing OSA methods along the time domain.

## 3 TIEM: TIME IMPORTANCE SCORE-AWARE EXTREMAL PERTURBATION MASK

### 3.1 OVERVIEW

We consider model-agnostic interpretation of video prediction models based on perturbation. In a video prediction model, a video with $T$ frames, each of width $W$ and height $H$, is considered as an input. The video is denoted by $\mathbf{X} = (\mathbf{x}_t)_{t=1}^T$, where $\mathbf{x}_t \in \mathbb{R}^{H \times W \times 3}$. The prediction model is denoted by $\Phi$, which can be either a classification model or a regression model. In the case of a regression model, it is expressed as $\Phi(\mathbf{X}) \in \mathbb{R}$, whereas in the case of a classification model, the probability of class $c$, when the ground truth class of $\mathbf{X}$ is $c$ out of the total $C$ classes, is expressed as $\Phi_c(\mathbf{X}) \in \mathbb{R}$ after passing through the softmax function. The objective of perturbation-based interpretation is to learn an importance map $\mathbf{M} = (\mathbf{m}_t)_{t=1}^T$, where $\mathbf{m}_t \in [0,1]^{H \times W}$, that visualizes the significant regions of $\mathbf{X}$ during prediction using $\Phi$. Each element of $\mathbf{m}_t$, $m_{i,j,t}$, corresponds to each pixel of frame $t$, $x_{i,j,t}$, where $i, j$ denote the spatial coordinates in each frame. The perturbation operation by the mask is expressed as $\mathbf{M} \otimes \mathbf{X} = \mathbf{M} \circ \mathbf{X} + (1 - \mathbf{M}) \circ (k * \mathbf{X})$, where $\circ$ denotes the Hadamard product, $*$ represents the convolution operator, and $k$ denotes a Gaussian blur kernel.

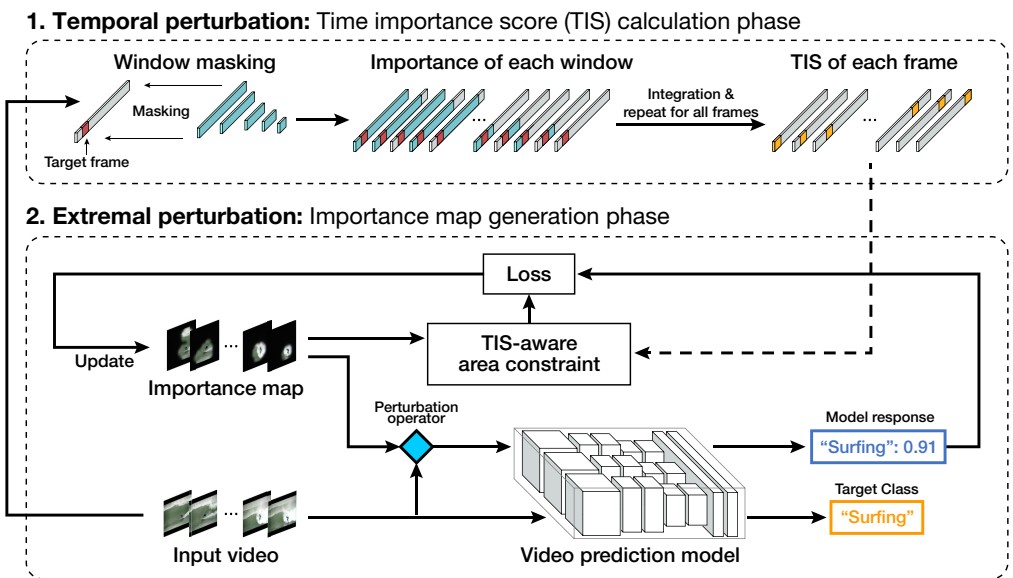

Figure 3: Flowchart of TIEM with temporal dynamics-focused dual perturbation. In the TIS calculation phase, the time importance score of each frame is evaluated considering different temporal contexts. In the importance map generation phase, the perturbation mask is fitted based on the score.

We propose a novel video interpretation method with temporal dynamics-focused dual perturbation, called time importance score-aware extremal perturbation mask (TIEM), which is illustrated in Fig. 3. TIEM addresses the challenges of addressing temporal dynamics in interpreting video predictions: temporal concentration and temporal spillover. To this end, we design the temporal

dynamics-focused dual perturbation approach that utilizes both temporal and extremal perturbations. It enhances the existing video interpretation methods by separating the learning structure for the importance map into two distinct dimensions: *temporal* and *spatial*. In a TIS calculation phase, TIEM appropriately calculates the temporal importance across frames, called the time importance score (TIS), using temporal perturbation. Then, in an importance map generation phase, based on the TIS, TIEM fits an importance map (i.e., the perturbation mask) that interprets the spatial domain of each frame, using extremal perturbation. This separate learning structure of TIEM allows it to overcome the challenges of video interpretation by avoiding over-integration of the spatial and temporal dimensions during the interpretation and by focusing on temporal dynamics more explicitly.

## 3.2 TEMPORAL PERTURBATION: CALCULATING TIME IMPORTANCE SCORES

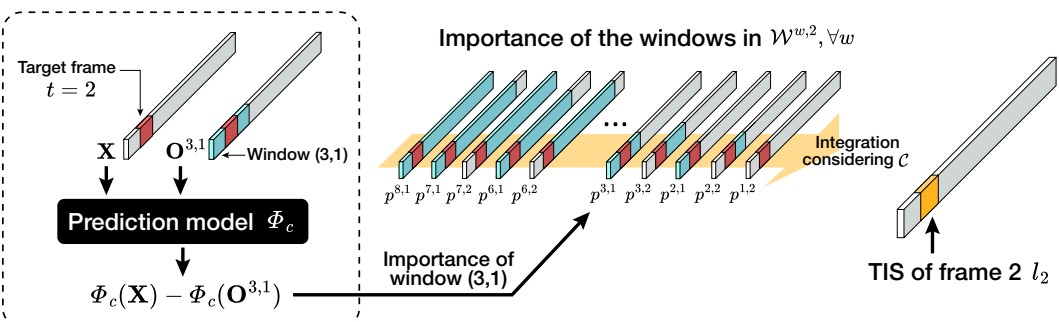

Figure 4: Conceptual illustration of the TIS calculation. The TIS of each frame $t$ is calculated by appropriately integrating the importance of the windows that contains frame $t$ (i.e., the set $\mathcal{W}^{w,t}, \forall w$).

In the TIS calculation phase, we introduce temporal perturbation to consider different temporal contexts. It refers to perturbing the video frame-by-frame via window masking with different window sizes. The importance of each frame is evaluated by analyzing the importance of windows of different lengths containing the frame and combining them as illustrated in Fig. 4. This allows us to evaluate the importance score of each frame in the time domain, focusing on temporal dynamics that comprehensively consider both short-term and long-term temporal contexts.

We denote a window of size $w$ starting at frame $t'$ by a pair of $(w, t')$. Then, the masked video by a window of $(w, t')$ is defined as

$$\mathbf{O}^{w,t'} = \left(\mathbf{o}_t^{w,t'}\right)_{t=1}^T, \text{ where } \mathbf{o}_t^{w,t'} = \begin{cases} \mathbf{0}^{H \times W \times 3}, & \text{if } t' \leq t < t' + w \\ \mathbf{x}_t, & \text{otherwise} \end{cases}. \quad (1)$$

In the masked video $\mathbf{O}^{w,t'}$, all frames in the interval from $t'$ to $t' + w$ are replaced by $\mathbf{0}^{H \times W \times 3}$ which is a zero tensor whose size is given by $H \times W \times 3$, indicating a black image. To calculate the TIS, masked videos are generated for all available different windows and inputted into the prediction model. Then, the difference in the model response of each masked video $\mathbf{O}^{w,t'}$ compared with the unmasked video $\mathbf{X}$ is computed as

$$p^{w,t'} = \frac{\Phi_c(\mathbf{X}) - \Phi_c(\mathbf{O}^{w,t'})}{\sum_{t'=1}^{T-w+1} p^{w,t'}}. \quad (2)$$

The difference $p^{w,t'}$ indicates the importance of the masked frames in $\mathbf{O}^{w,t'}$ for the model prediction. The set of the importance of windows is given by $\mathcal{P} = \{\mathbf{p}^w\}_{w=1}^T$, where $\mathbf{p}^w = \{p^{w,t'}\}_{t'=1}^{T-w+1}$, considering various lengths of temporal context.

Since the length of the temporal context required for accurate interpretation may vary, we choose valid window sizes to consider in the calculation of TIS, rather than using all window sizes. The set of valid window sizes is given by

$$\mathcal{C} = \{w | \theta^w > \alpha \cdot \max(\boldsymbol{\theta})\}, \quad (3)$$

where $\alpha < 1$ is a hyperparameter that controls the sensitivity of the filtering, $\boldsymbol{\theta} = (\theta^w)_{w=1}^T$, and $\theta^w = \sum_{t'=1}^{T-w+1} \left| \frac{dp^{w,t'}}{dt'} \right|$. This filtering process utilizes the total variation $\theta^w$ in the temporal changes

of window $w$ in the set $\mathcal{P}$, which enables more accurate interpretation of videos with relatively high frame-to-frame variation rates. We define a set of the windows of size $w$ containing frame $t$ by $\mathcal{W}^{w,t} = \{(w,t')|t \in [t', t'+w)\}$. Based on the importance of windows and valid window sizes, we calculate the TIS of frame $t$ as

$$l_t = \frac{\sum_{w \in \mathcal{C}} \sum_{(w,t') \in \mathcal{W}^{w,t}} p^{w,t'}}{\sum_{w \in \mathcal{C}} \sum_{(w,t') \in \mathcal{W}^{w,t}} 1}. \tag{4}$$

This TIS calculation is carried out by accumulating the importance of the valid windows. Since the number of valid windows may differ across the frames, the TIS is divided by the number of valid windows to scale the differences. The TIS vector across the frames is given by $\mathbf{l} = (l_t)_{t=1}^T$. For ease of presentation, we assume that after the TIS calculation, the TIS vector is normalized so that its sum is 1 as $\mathbf{l}(\sum_{t=1}^T l_t)^{-1}$. The pseudo-code of the TIS calculation is provided in the appendix.

### 3.3 Extremal Perturbation: Generating TIS-Aware Importance Map

In the importance map generation phase, a TIS-aware importance map is fitted based on extremal perturbation. We introduce an important region ratio $a$, which is a hyperparameter that represents the ratio of the region preserved. The size of the important region of each frame is constrained individually by using both $a$ and TIS $\mathbf{l}$ together, unlike the existing methods that constrain the total size of the region over the expanded spatio-temporal dimension using $a$. More specifically, a TIS-aware importance map with the ratio $a$ can be obtained as

$$\mathbf{M}_{\mathbf{l}_a} = \underset{\mathbf{M}:\|\mathbf{m}_t\|_1 = al_t THW, \forall t}{\arg\max} \Phi_c(\mathbf{M} \otimes \mathbf{X}), \tag{5}$$

where $\|\cdot\|_1$ denotes the $L_1$ norm. The TIS-aware importance map, $\mathbf{M}_{\mathbf{l}_a}$, restricts the region size of each frame $t$ in the video according to the corresponding TIS-aware ratio of frame $t$, $a \cdot l_t$. Furthermore, to take advantage of extremal masks (Fong et al., 2019; Li et al., 2021), we find the smallest TIS-aware importance map that achieves the lowest baseline bound $\Phi_0$ based on the smallest important region ratio $a^*$, defined as

$$a^* = \min\{a : \Phi_c(\mathbf{M}_{\mathbf{l}_a} \otimes \mathbf{X}) \geq \Phi_0\}. \tag{6}$$

It is worth emphasizing that the extremal perturbation in equation 6 is different from those in Fong et al. (2019); Li et al. (2021), since its importance map considering the TIS across the frames.

To solve equation 5 using typical gradient-based methods, we construct a loss function that regularizes the TIS-aware important region for each frame as follows:

$$\mathbf{M}_{\mathbf{l}_a} = \underset{\mathbf{M}}{\arg\min}\{\lambda \sum_{t=1}^T \|\text{vecsort}(\mathbf{m}_t) - \mathbf{r}_{al_t}\|^2 - \Phi_c(\mathbf{M} \otimes \mathbf{X})\}, \tag{7}$$

where $\lambda$ is a hyperparameter for the region regularization and $\mathbf{r}_{al_t}$ is the region regularization vector that consists of $al_t HW$ ones followed by $(1 - al_t)HW$ zeros. In equation 7, the TIS-aware area constraint for each frame is considered individually as a regularization term, thereby independently constraining the regions of each frame. This ensures that the important region of each frame can be fitted focusing on the key factors without temporal concentration and temporal spillover.

## 4 Experimental Results

In this section, we evaluate the performance of our proposed method, TIEM, through experiments. First, we consider a synthetic white-box regressor based on video data, whose internal process of how the model predicts is perfectly known. As illustrated in Fig. 2, interpreting the white-box model allows us to clearly evaluate how well each visual explanation method performs. Furthermore, we interpret a black-box model for real-world videos by using visual explanation methods. This allows us to visually verify whether each method provides a plausible interpretation of the black-box model for real-world videos. In the experiments, we consider the model-agnostic visual explanation methods, EP-3D and STEP, for comparison. Both methods are based on perturbation masks as in TIEM. EP-3D is a method that simply extends the spatial dimension to the spatio-temporal dimension, while STEP is a method that additionally considers smoothing the perturbation masks across adjacent frames.

## 4.1 WHITE-BOX REGRESSOR

We consider a simple white-box regressor whose predictions only rely on a known target subset $\mathcal{A}$ of the entire pixels composed of a video. Each pixel of the video can be indicated by a tuple $(t, i, j)$, where $t \in \{1, \cdots, T\}$ denotes the frame number and $(i, j) \in \{1, \cdots, H\} \times \{1, \cdots, W\}$ denote the spatial coordinates. Therefore, the target subset $\mathcal{A}$ contains the tuples whose corresponding pixels are used for the regressor. In a mathematical expression, for a given video $\mathbf{X} = (\mathbf{x}_t)_{t=1}^T$, where $\mathbf{x}_t \in \mathbb{R}^{H \times W}, \forall t$, the white-box regressor is defined as

$$f(\mathbf{X}) = \sum_{(t,i,j) \in \mathcal{A}} (x_{t,i,j})^2. \tag{8}$$

Since the output of the regressor depends only on the pixels belonging to the target subset $\mathcal{A}$, the ground truth of its visual explanation is the region composed of these pixels.

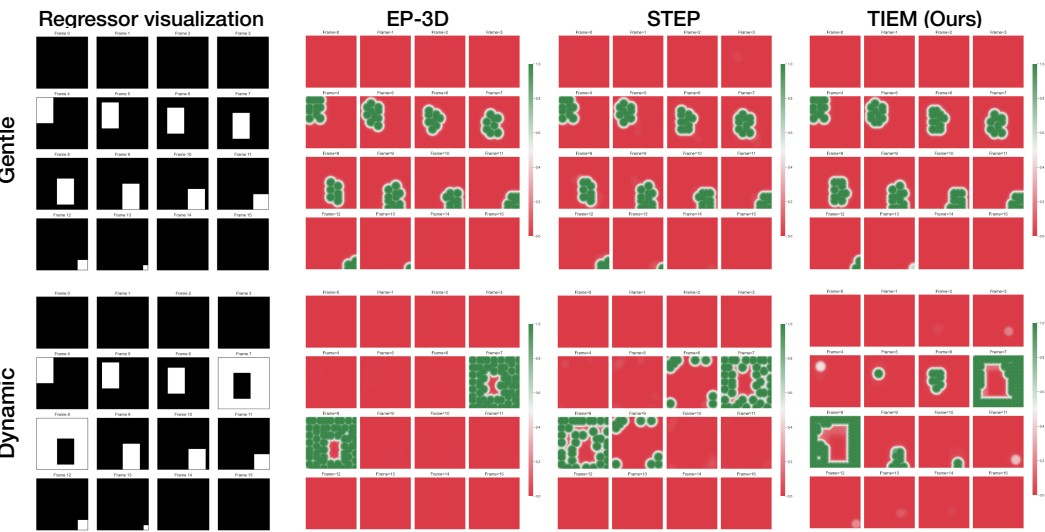

Figure 5: The experimental results of the white-box regressor. The white regions indicate the pixels used for the regressor, thereby being the ground truth of its visual explanation. The frame numbers start from 0 and increase from left to right, top to bottom.

To clearly show the performance of the methods, we consider two regressors whose target subsets change gently and dynamically across frames, respectively. The regressors are visualized in Fig. 5 by highlighting the target subset in white. In the gentle regressor's visualization at the top of the figure, the white rectangle gradually moves diagonally downward. This represents a gentle video in which the significant regions across frames change gradually in the spatial domain. In the dynamic regressor's visualization at the bottom of the figure, the white rectangle moves identically to the gentle one, but in frames 7 and 8, the white regions are momentarily reversed (i.e., the outside of the black rectangle becomes a target subset). This represents a dynamic video in which the significant regions change rapidly or appear ephemerally across frames.

In the figure, the three images to the right side of each regressor visualization show the visual explanation results for the white-box regressor by using EP-3D, STEP, and our method. For a fair comparison, all methods are configured to highlight up to 10% of the regions (i.e., the area constraint of each method is set to be 10%). Ideally, the green regions should lie perfectly within the white regions of the regression visualization.

We first investigate the results from the qualitative perspective. In the results of the gentle regressor, the visual explanation of the three methods is quite similar, effectively identifying the consistently moving white rectangle. On the other hand, in the results of the dynamic regressor, the visual explanation of the three methods is significantly different. First, in the visual explanation of EP-3D, the temporal concentration occurs, where the important region is overwhelmingly concentrated in the crucial frames (i.e., frames 7 and 8). As a result, the white regions in other frames are completely neglected. In the visual explanation of STEP, although the temporal concentration is not as severe as

in EP-3D, the temporal spillover occurs in frames 6 and 9 as also described in Section 1.1. Finally, TIEM identifies the white rectangles in frames 6 and 9, while EP-3D and STEP fail to do so. It also detects, albeit minimally, the white rectangles in frames 4, 5, 9, and 10.

To quantitatively evaluate the performance of the methods, we introduce a pointing game as a metric and apply it to the results in Fig. 5. The idea of the pointing game is to assess how well the visual explanation identifies the important region of a video (Petsiuk et al., 2018). In the pointing game for the white-box regressors, the accuracy of the visual explanation is quantified as

$$\text{Pointing Game (\%)} = \frac{\text{Highlighted region in } \mathcal{A} - \text{Highlighted region out of } \mathcal{A}}{\text{Total highlighted region}} \times 100.$$

This metric explicitly reflects the temporal spillover, considering the highlighted region out of $\mathcal{A}$. If the highlighted region of a method perfectly lies within the white region in the regressor visualization, its pointing game score becomes 100%. On the other hand, if none of the highlighted regions lies within the white region, its pointing game score becomes -100%.

We provide the pointing game scores for the white-box regressors in Table 2. For the gentle regressor, all three methods achieve comparable results of about 90%, as their visual explanations are similar as shown in Fig. 5. For the dynamic regressor, TIEM significantly outperforms the other methods, achieving nearly 100%. This implies that almost every region highlighted by TIEM lies within the target subset, as shown in Fig. 5. EP-3D achieves about 91% as its highlighted regions are highly

Table 2: Pointing Game for White-Box Regressor (%)

|      | Gentle      | Dynamic     |
|------|-------------|-------------|
| EP-3D | 89.99±0.21 | 91.85±0.27 |
| STEP  | 89.68±1.05 | 65.35±1.39 |
| TIEM  | 90.14±0.28 | **99.79±0.06** |

concentrated in frames 7 and 8. STEP achieves only about 65% since it highlights the regions out of the target subset due to the temporal spillover.

These results demonstrate that the separate learning structure of TIEM via dual perturbation qualitatively and quantitatively outperforms the existing methods. It effectively mitigates the two key challenges of the existing methods–temporal concentration and temporal spillover. As a result, TIEM can be effectively applied not only to gentle videos but also to dynamic videos.

## 4.2 BLACK-BOX CLASSIFIER FOR REAL-WORLD VIDEOS

The state-of-the-art models for video classification are highly complex and diverse. We here consider an R(2+1)D model (Tran et al., 2018), a widely used 3D-CNN architecture, as an action recognition model to compare the interpretability of the methods. Specifically, we fine-tune the R(2+1)D-18 architecture that is pretrained on the Kinetics-400 dataset for our experiments. For comparison, we use the UCF101-24 dataset (Soomro, 2012) which is a large-scale video dataset widely used for video-based learning such as action recognition and video classification. It includes 101 action classes, covering a variety of sports, exercises, and daily activities. The dataset contains a total of 13,320 video clips, each extracted from real videos collected from the internet. To focus on temporal dynamics from the object's movements, we customize the UCF101-24 dataset by adding videos of swimming strokes, in which key movements of each swimming stroke appear ephemerally during a specific frame segment. The details of the hyperparameters of the model and more experimental results with another video classification model and other samples are provided in the appendix.

We provide a visual explanation of the methods for the action recognition model in Fig. 6. In the figure, each video represents the sample fed into the model for prediction. We consider two video samples of swimming strokes, "front crawl" at the top of the figure and "breaststroke" at the bottom of the figure, which have different temporal dynamics. For each video, some frames show the movement of the corresponding swimming stroke that is distinct from other types of strokes. We call such frames *signature* frames and highlight them in red. Therefore, the type of swimming stroke in the video can be recognized by its signature frames and it is difficult to distinguish the type of swimming stroke in the video by examining only the non-signature frames that do not contain any distinctive movement. In the front crawl video, the distinctive movement of the front crawl, where the arms move in a crossing pattern with a leg kick, begins at frame 7. In the breaststroke video, the distinctive movement of the breaststroke, where the body lifts for breathing, the circular arm stroke, and the frog-like leg kick come together, appears in the segment from frame 5 to frame 9.

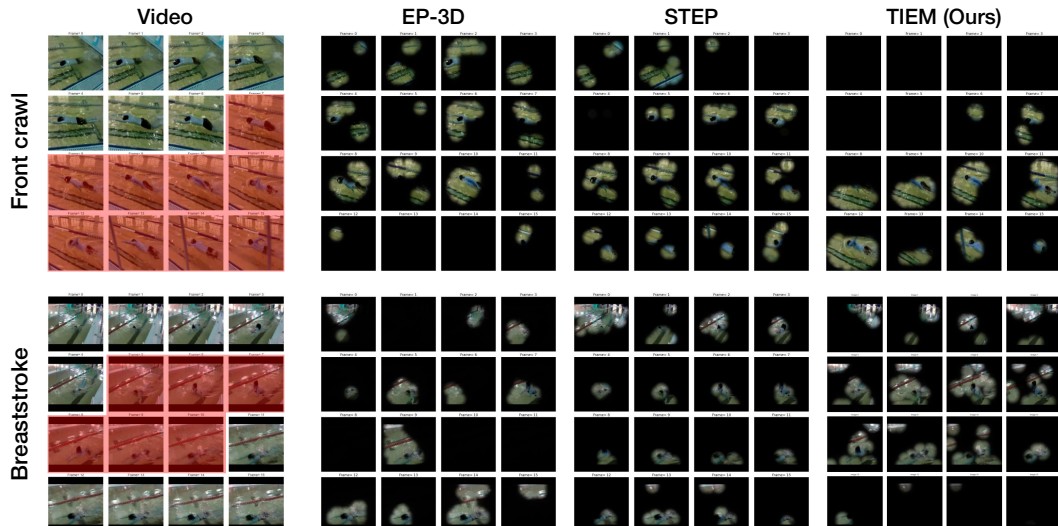

Figure 6: The experimental results of the black-box model for real-world videos. In the videos, the signature frames highlighted in red contain the distinctive movements of the corresponding stroke. The frame numbers start from 0 and increase from left to right, top to bottom.

In the figure, the three images on the right side of each video show the visual explanation results of the video using EP-3D, STEP, and our TIEM. Each visual explanation interprets the prediction from the action recognition model by visualizing the important region appearing as it is while the unimportant region is masked in black. As in the white-box regressor experiments, the area constraint of each method is set to be 10% for fair comparison.

We first examine the results of the front crawl video. EP-3D effectively extracts the spatial information of the swimmer from each frame. However, in a temporal aspect, its importance map appears to focus on frames non-signature frames 0–6 rather than the signature frames (especially, frames 12–15). This indicates that EP-3D is ineffective in extracting the temporal dynamics of the front crawl video. STEP is more effective than EP-3D in considering the temporal dynamics and its importance map tracks the swimmer well across the frames. However, it still fails to focus on the key temporal region (i.e., the signature frames). In contrast, our TIEM effectively focuses on the key temporal region, extracting the spatial information in the signature frames.

In the results of the breaststroke video, EP-3D and STEP still fail to focus on the signature frames, and particularly they concentrate their importance map on a few specific frames. EP-3D did not extract any spatial information at all for signature frames 8 and 10. In STEP, the spatial information of the signature frames is partially unmasked as smoothing it, but the majority of its importance map appears outside the signature frames. On the other hand, TIEM not only extracts significant spatial information from each frame well, but also effectively focuses on the signature frames when generating an importance map. As a result, its importance map clearly unmasks the swimmer's movement in the signature frames. In real-world videos, it is challenging to explicitly assess temporal concentration and spillover due to the nature of the model and the ambiguity of the videos, which makes the ground truth unclear. Nevertheless, these results demonstrate that TIEM significantly mitigates a limitation of the existing methods in focusing their analysis on frames outside the signature frames.

To quantify how well each method considers the temporal dynamics in videos, we design a temporal pointing game as a metric. Contrary to the pointing game in Section 4.1, it focuses on the time domain (i.e., the signature frames of the target video) since the ground truth of the visual explanation in real-world videos is unclear. In the temporal pointing game for the black-box classifier, the temporal accuracy of the importance map is evaluated as

$$\text{Temporal Pointing Game (\%)} = \frac{\text{Unmasked region in the signature frames}}{\text{Total unmasked regions}} \times 100.$$

If the unmasked region of a method perfectly belongs to the signature frames, its temporal pointing game score becomes 100%. On the other hand, if it perfectly belongs to frames outside the signature frames, the result becomes 0%.

We provide the temporal pointing game scores for the black-box classifier with two videos in Table 3. From the results, we can see that our TIEM significantly outperforms the other two methods in terms of considering the temporal dynamics. In particular, for the front crawl video, TIEM achieves a result close to 100%. On the other hand, EP-3D achieves the result of around 47%, while STEP is superior to EP-3D by about 8% due to its tendency to continuously detect objects. It

Table 3: Temporal Pointing Game for Black-Box Classifier (%)

|  | **Front crawl** | **Breaststroke** |
|---|---|---|
| EP-3D | 47.57±5.38 | 24.52±1.38 |
| STEP | 55.32±9.92 | 32.53±1.98 |
| TIEM | **98.73±0.17** | **65.22±0.34** |

is difficult to say that EP-3D and STEP examine the temporal dynamics well, considering that the signature frames account for 56.25% of the video. For the breaststroke video, the temporal pointing game scores of all methods decrease compared with those for the front crawl video due to the shorter segment of the signature frames. TIEM achieves a score of about 65%, which is quite larger than the given ratio of the signature frames to the video of 37.5%, while EP-3D and STEP achieve scores of about 24–32%. This shows that EP-3D and STEP do not effectively consider temporal dynamics in their visual explanation.

These results demonstrate that TIEM outperforms the existing model-agnostic perturbation-based methods in terms of interpreting the black-box model for real-world videos. In particular, TIEM effectively identifies the signature frames compared with the existing methods, focusing on the temporal dynamics of videos by its dual perturbation process. This shows that TIEM can be used for real-world applications, where an action is presented ephemerally.

## 4.3 DISCUSSIONS

The proposed method, TIEM, can effectively interpret video predictions via its temporal dynamics-focused dual perturbation. The experimental results demonstrate this strength, but TIEM can still be enhanced in a variety of aspects. In particular, we observe a few limitations of TIEM in the results.

The process for calculating the TIS can be improved and more sophisticated. In TIEM, the average importance of the windows to which a frame belongs is considered as the TIS of the frame as described in equation 4. This is a simple and effective way to represent the temporal importance of each frame, but it makes limited use of the given windows. For example, an insignificant frame may be overestimated if the frame is included in a long window containing significant frames as in frame 1 of the breaststroke video in Fig. 6. (The detailed analysis of the TIS is provided in the appendix.) To avoid this, a more powerful temporal explanation of frames can be synthesized by considering more complex connections between frames through the interaction of multiple windows.

The TIS-aware importance map can be generated more precisely. TIEM fits the importance map using the TIS of each frame. This effectively considers the temporal information across frames, but it does not address spatial information appearing across frames continuously as shown in Fig. 6. As a result, TIEM focuses well on the signature frames, but often extracts spatial information that is discontinuous across the frames. Therefore, if we extend its importance map generation process to exploit the spatial information of adjacent frames together, such as STEP, a more complete spatio-temporal perturbation mask can be obtained.

## 5 CONCLUSION

In this paper, we have studied model-agnostic visual explanations for videos via perturbation. We have first examined the key challenges of the existing methods–specifically, temporal concentration and temporal spillover–which are especially prevalent in dynamic videos. To address these issues, we proposed a novel method called TIEM, which utilizes a dual perturbation strategy focused on temporal dynamics. This method incorporates temporal perturbation to evaluate the TIS across frames in a video and extremal perturbation to generate a TIS-aware importance map for the video. The dual perturbation strategy enables TIEM to effectively capture the temporal dynamics within a video by explicitly exploiting the TIS of a video when fitting the importance map for the video. Our experiments with synthetic and real-world videos demonstrated that TIEM outperforms the existing methods. In particular, it is clearly shown that the concept of dual perturbation in TIEM mitigates the key challenges of the existing methods.

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

## A  PSEUDOCODE OF TIS CALCULATION

The pseudocode of the TIS calculation with temporal perturbation is presented in Algorithm 1.

---

**Algorithm 1** Calculate Time Importance Score

---

1: **Input:** Original input $\mathbf{X} = \{\mathbf{x}_t\}_{t=1}^{T}$, $\mathbf{x}_t \in \mathbb{R}^{H \times W}$, Model $\Phi_c$, Threshold ratio $\alpha$
2: **Output:** TIS l
3: Initialize an empty array $\mathbf{O}, \mathcal{P}, \mathcal{C}, \boldsymbol{\theta}, \mathbf{l}, \boldsymbol{\lambda} = (\lambda_t)_{t=1}^{T}$
4: $\boldsymbol{\lambda} \leftarrow \mathbf{0}^T$
5: **for** $w = 1$ **to** $T$ **do**
6:     **for** $t = 1$ **to** $T - w + 1$ **do**
7:         $\mathbf{O}^{w,t} \leftarrow \mathbf{X}$
8:         **for** $t' = t$ **to** $t + w$ **do**
9:             $\mathbf{o}_{t'}^{w,t} \leftarrow \mathbf{0}^{H \times W \times 3}$
10:         **end for**
11:         $p^{w,t} = \Phi_c(\mathbf{X}) - \Phi_c(\mathbf{O}^{w,t})$
12:     **end for**
13:     $\mathbf{p}^w = \mathbf{p}^w/\mathrm{sum}(\mathbf{p}^w)$
14: **end for**
15: **for** $w = 1$ **to** $T$ **do**
16:     $\theta^w = \sum_{t=1}^{T-w+1} \left| \frac{dp^{w,t}}{dt} \right|$
17: **end for**
18: $\mathbf{C} = \{w \mid \theta^w > \alpha \cdot \max(\boldsymbol{\theta})\}$
19: **for** $w$ **in C do**
20:     **for** $i = 1$ **to** $\mathrm{len}(\mathbf{p}^w)$ **do**
21:         **for** $t' = i$ **to** $i + w$ **do**
22:             $l_{t'} += p^{w,t'}$
23:             $\lambda_{t'} += 1$
24:         **end for**
25:     **end for**
26: **end for**
27: $\mathbf{l} = \left( \frac{1}{\boldsymbol{\lambda}} \right) / \mathrm{sum} \left( \frac{1}{\boldsymbol{\lambda}} \right)$

---

## B  DETAILS OF EXPERIMENTAL SETUP

Table 4: R(2+1)D-18 Model parameter

| Type | In/Out | Kernel |
|---|---|---|
| Conv3d | $(3, 16, 128, 128)/(64, 16, 64, 64)$ | $(1 \times 7 \times 7)$ |
| BatchNorm3d | $(64, 16, 64, 64)$ | - |
| ReLU | $(64, 16, 64, 64)$ | - |
| MaxPool3d | $(64, 16, 64, 64)/(64, 16, 32, 32)$ | $(1 \times 3 \times 3)$ |
| 2 x R(2+1)D Block | $(64, 16, 32, 32)/(64, 16, 32, 32)$ | $(1 \times 3 \times 3), (3 \times 1 \times 1)$ |
| 2 x R(2+1)D Block | $(64, 16, 32, 32)/(128, 8, 16, 16)$ | $(1 \times 3 \times 3), (3 \times 1 \times 1)$ |
| 2 x R(2+1)D Block | $(128, 8, 16, 16)/(256, 4, 8, 8)$ | $(1 \times 3 \times 3), (3 \times 1 \times 1)$ |
| 2 x R(2+1)D Block | $(256, 4, 8, 8)/(512, 2, 4, 4)$ | $(1 \times 3 \times 3), (3 \times 1 \times 1)$ |
| AdaptiveAvgPool3d | $(512, 2, 4, 4)/(512, 1, 1, 1)$ | - |
| FC1 | $(512)/(400)$ | - |
| FC2 | $(400)/(128)$ | - |
| FC3 | $(128)/(28)$ | - |

In this section, we provide a more detailed explanation of the video prediction model architecture and the hyperparameters used for training, which were not sufficiently covered in the experimental setup described in Section 4.2 and Appendix C. First, for the R(2+1)D model included in the main paper, the model was constructed in the PyTorch environment as shown in Table 4, and we utilized

a pretrained model on the Kinetics-400 dataset for transfer learning. The model was trained using the Adam optimizer with a learning rate of 1e-3 for 50 epochs. The trained model achieved 100% accuracy on video classification within the dataset split into 70% for training, 9% for testing, and 21% for validation.

Table 5: ResNet-50 LSTM Model Parameters

| Type | In/Out | Kernel |
|---|---|---|
| Conv2d | $(3, 128, 128)/(64, 64, 64)$ | $7 \times 7$ |
| BatchNorm2d | $(64, 64, 64)$ | - |
| ReLU | $(64, 64, 64)$ | - |
| MaxPool2d | $(64, 64, 64)/(64, 32, 32)$ | $3 \times 3$ |
| ResNet Layer3 | $(512, 16, 16)/(1024, 8, 8)$ | - |
| ResNet Layer4 | $(1024, 8, 8)/(2048, 4, 4)$ | - |
| AdaptiveAvgPool2d | $(2048, 4, 4)/(2048, 1, 1)$ | - |
| FC1 | $(2048)/(1024)$ | - |
| FC2 | $(1024)/(512)$ | - |
| LSTM | $(512)/(256)$ | - |
| FC3 | $(256)/(28)$ | - |

In Appendix D, additional interpretation experiments were conducted using the ResNet50-LSTM (R50LSTM) model to demonstrate the model-agnostic characteristic of the proposed TIEM method. The parameters of the trained model are shown in Table 5. The model was structured so that the final FC2 layer of the ResNet50 model, which receives a single frame as input, was connected to the input of the LSTM model at each timestep. This model was also constructed within the PyTorch environment, and we utilized a pretrained model on the ImageNet dataset (Deng et al., 2009) for transfer learning. The Adam optimizer was used for training with a learning rate of 5e-5 over 100 epochs. Since the main objective of this experiment was not to improve model performance but to evaluate the proposed visual explanation method, hyperparameters were fine-tuned for this purpose. The model achieved relatively high classification accuracies of 92.8% on the validation dataset and 95.2% on the test dataset, using the same dataset split as in the previous experiment.

## C  TIS ANALYSIS OF EXPERIMENTAL RESULTS IN SECTION 4.2

In this section, we provide an analysis of the TIS utilized in the calculation of the importance maps using the TIEM for the "front crawl" and "breaststroke" videos, as discussed in Section 4.2.

In Fig. 7 and Fig. 8, we provide the TIS of each frame and the importance of each window size in two videos calculated by TIEM. In Section 4.2, when analyzing the overall results by focusing on the signature frames that were crucial in the temporal sequence of each video, it is evident that the TIEM method effectively detects the key temporal regions of the video. The results are based on the calculated TIS for each video, which can be seen in Fig. 7a and Fig. 8a.

In Fig. 7b, when examining the importance of front crawl video across different window sizes ranging from $w = 1$ to $w = 4$, which captures relatively short-term temporal context, the early part of the video is highlighted as important. Following this, as longer-term temporal context is considered, we observe an increase in importance for the later part of the video. This indicates that analysis results may vary depending on the extent to which temporal context between frames is accounted for, illustrating that relying on the temporal context of a single frame to calculate temporal importance can hinder accurate analysis of the overall video's impact on the prediction model.

Furthermore, the window size that recorded the highest peak among all windows is $w = 5$. At $t = 8$ of $w = 5$, the TIS recorded a score close to 1, where the window masks the video frames 8-13. This nearly coves the key temporal region of the signature frame, excluding just one frame on each side. This result indicates that the key temporal region is almost perfectly identified by TIS.

Fig. 7a is computed based on the importance of each window size from Fig. 7b. In this experiment, $\alpha$ was set to 0.8, and the TIS was calculated using the valid window sizes $\mathcal{C} = \{5, 6, 7\}$.

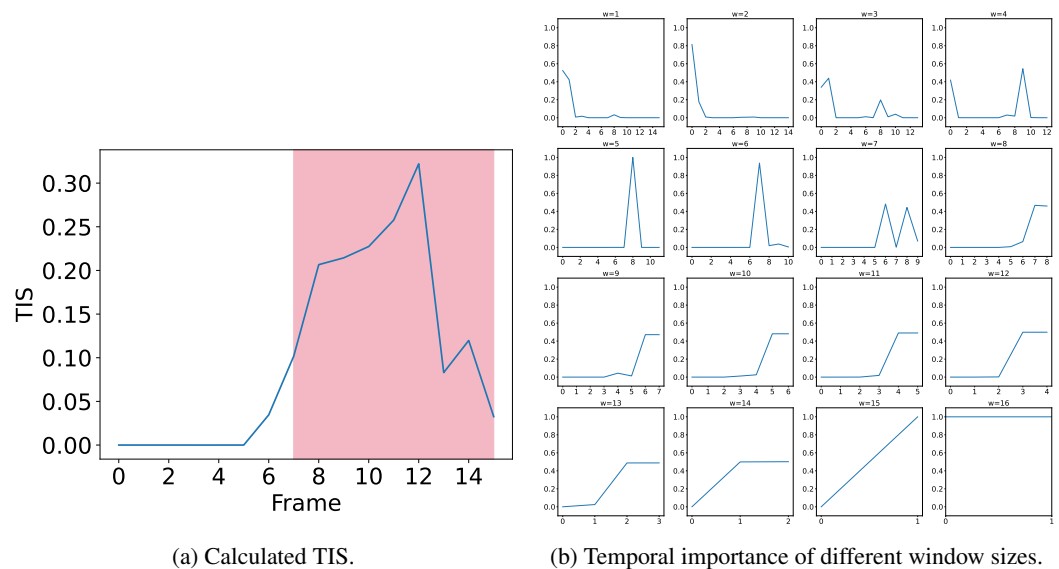

(a) Calculated TIS.

(b) Temporal importance of different window sizes.

Figure 7: TIS analysis results of the front crawl video. In the temporal importance results, the window numbers start from 1 and increase from left to right, top to bottom.

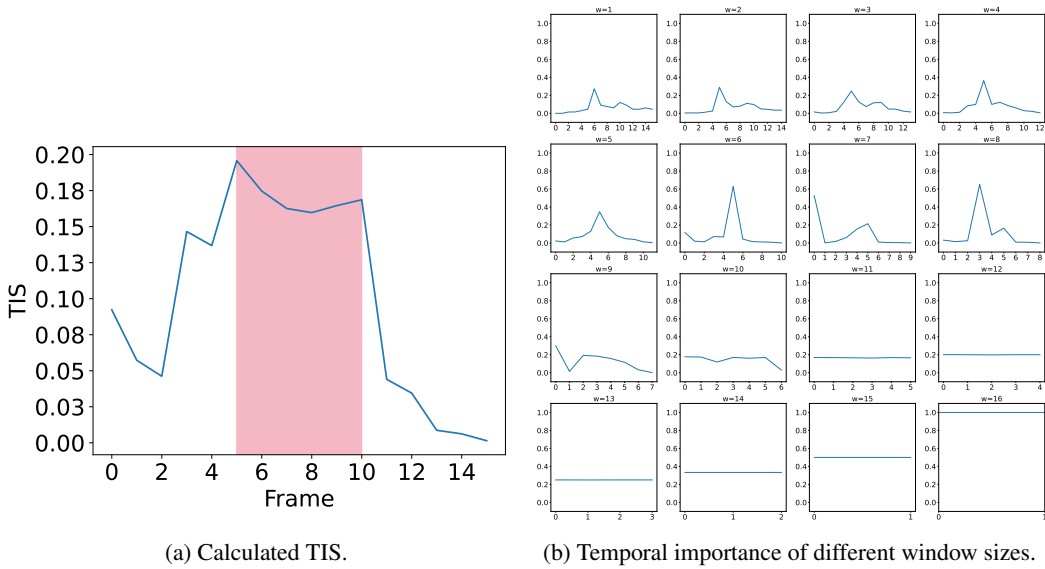

(a) Calculated TIS.

(b) Temporal importance of different window sizes.

Figure 8: TIS analysis results of the breaststroke video. In the temporal importance results, the window numbers start from 1 and increase from left to right, top to bottom.

Next, in the case of Fig. 8b, unlike Fig. 7b, most of the importances were concentrated in the middle section of the video across the different window sizes, showing consistent behavior. The highest peaks were observed for $w = 6$ and $w = 8$, which regions also recorded high scores at smaller window sizes. At $w = 6$, the peak occurs at $t = 5$, and at $w = 8$, the peak occurs at $t = 3$, with frames 5-11 and frames 3-11 being masked, respectively. Both regions include the key region of signature frames from 5 to 10. The TIS for Fig. 8a was calculated using the same $\alpha = 0.8$ as the previous video, and the valid window size was $\mathcal{C} = \{6, 8\}$.

Through the TIS analysis of the two preceding videos, we confirmed that different window sizes may need to be considered for accurate temporal context analysis of the videos, and we can observe the calculation process of TIS, which successfully identifies the key temporal regions.

## D  ADDITIONAL EXPERIMENTAL RESULTS

In this section, we additionally present the interpretation results of the R50LSTM model, as discussed in Appendix B.

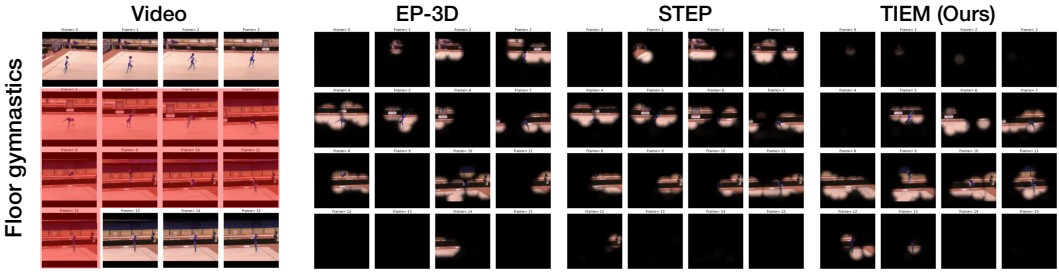

Figure 9: The experimental results of the black-box model for the floor gymnastics video. In the videos, the signature frames highlighted in red contain the distinctive movements of the corresponding stroke. The frame numbers start from 0 and increase from left to right, top to bottom.

In Fig. 9, we provide a visual explanation of the methods using the R50LSTM model with CNN-LSTM structure, to demonstrate the model-agnostic characteristic of TIEM. In contrast, Section 4.2 employed the R(2+1)D model with a 3D-CNN structure. The video shown in the figure depicts a "floor gymnastics" action, and as in previous experiments, we highlight the frames where the gymnastics occur in red, assigning them as signature frames, while frames before and after, where the athlete is standing, are assigned as non-signature frames.

Analyzing the interpretation results of each method, all three methods exhibit the same patterns observed in the previous experiments with the R(2+1)D model. First, in the case of EP-3D, although the method effectively extracted the spatial information of the gymnast in each frame, it highlighted non-signature frames 1-3 and 14 from a temporal aspect. Even within the signature frames, it detected the gymnast in a discontinuous form rather than continuously emphasizing the appearance of a gymnast.

In the case of STEP, it detected the gymnast more continuously within the signature frames compared to EP-3D. However, this method also highlighted the non-signature frames 1-3. Both techniques, similar to previous experiments, sporadically identified the spatial information of each frame but failed to locate the key temporal regions. In contrast, TIEM effectively focused on the key temporal regions, and the spatial information captured in each frame was connected continuously.

Table 6: Temporal Pointing Game for Black-Box Classifier (%)

|  | **Floor gymnastics** |
| --- | --- |
| EP-3D | 62.54±9.76 |
| STEP | 57.42±4.68 |
| TIEM | **99.89±0.25** |

The fact that TIEM focused on key temporal regions more effectively than the other methods is also evident in Table 6, which is calculated using the same formula as Table 3. While EP-3D recorded a score in the 60% range and STEP achieved a score of 57%, TIEM showed results close to 100%. This suggests that TIEM reflects temporal dynamics more effectively than the other two methods, successfully focusing on the signature frames.

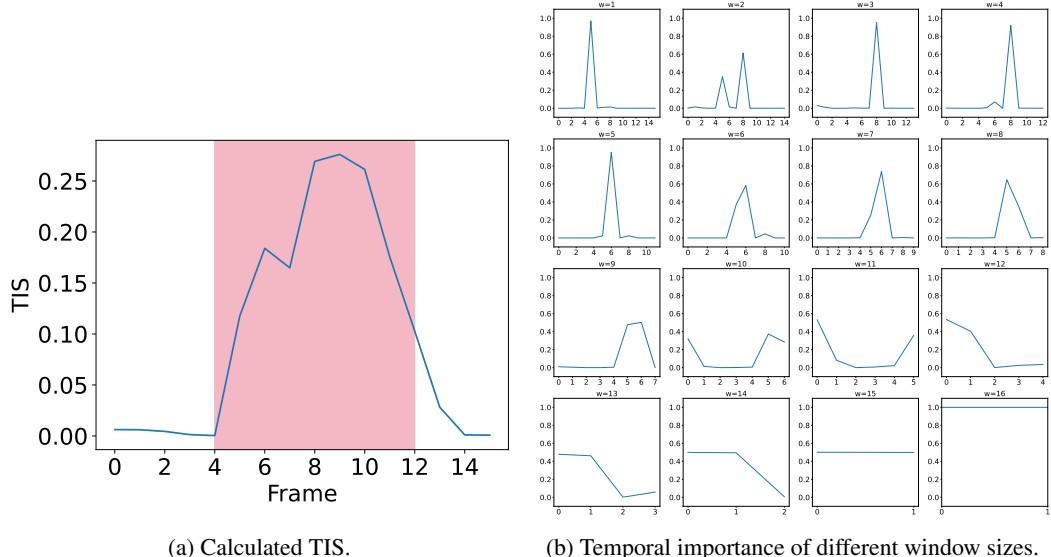

(a) Calculated TIS.

(b) Temporal importance of different window sizes.

Figure 10: TIS analysis results of the floor gymnastics video. In the temporal importance results, the window numbers start from 1 and increase from left to right, top to bottom.

Fig. 10 visualizes the TIS and importance of each window size, calculated by using TIEM in the additional experiment. First, in Fig. 10b, for the cases of $w = 1$ and $w = 2$, which consider relatively short-term temporal context, we can observe that frame 5, located at the beginning of the signature region, was identified as an important frame. As $w$ increases, the latter part of the signature region becomes more influential, with frames 8-12 being identified as the most influential when $w = 4$. As $w$ continues to grow, considering long-term dependencies, when the signature region is either absent or only slightly included, frames from the non-signature region, especially in the earlier parts, are found to have a greater impact on the model's prediction.

Based on these window-specific TIS values, Fig. 10a was computed with $\alpha = 0.6$ and $\mathcal{C} = \{1, 2, 3, 4, 5, 6, 7, 8\}$. As a result, it can be observed that the TIS of the videos was analyzed using a small window size that considers a relatively short-term temporal context.

