# OpenReview forum: "TIEM: Enhancing Explanation of Video Prediction via Temporal Dynamics-Focused Dual Perturbation"
_ICLR.cc/2025/Conference — ICLR 2025 Conference Withdrawn Submission_

### Official Review · Reviewer_FjMq · 2024-11-01

**Soundness:** 2
**Presentation:** 2
**Contribution:** 2
**Rating:** 3
**Confidence:** 3

**Summary:**

This work introduces a framework for the explainability of video prediction models. It highlights two issues observed from applying existing approaches to the video setting: “temporal concentration” and “temporal spillover”. The first refers to emphasising only very few video frames as significant for model prediction; the second comes from the explanation having a strong temporal rather than content prior.
These two observations lead to development of a perturbation-based approach to address the issues. The approach has two steps. In the first step, the approach estimates an importance score for each frame by removing the frame (and some of its adjacent frames) from the video and noting the difference in the model prediction. The second step integrates the importance scores to estimate spatial masks. The idea is that perturbing the frame with the mask should have a maximal effect on the model output while the overall budget of the available perturbation area is limited in proportion to the importance scores.
The experiments on two synthetic and two real video sequences show more temporally localised attribution maps.

**Strengths:**

The text discusses some limitations of previous work and provides clear motivation for the approach with an illustration of the issues on a toy example.

The approach is interesting and straightforward to implement.

The readability of the technical aspects is good, on a high level (please see below for details); the accompanying illustrations smoothly guide the reader through the technical exposition.

**Weaknesses:**

The reader is likely to stumble right at the beginning: the work does not pose the problem in a sufficiently clear way. There is a whole zoo of video predictive models, which depend on a specific task. It is not clear why there should be a single explanation, as the work implicitly assumes, and what this explanation should be expected to convey. I would encourage the authors to mediate on the specific video task, its output representation and behaviour on video sequences. It is not clear how the proposed approach can be used a xAI tool to explain the predictions of (any) video model.

The experiments are only scratching the surface. The experiment in Fig. 2 is okay for problem illustration (though I’m again not 100% sure what explanation we expect here, see the previous point). However, this synthetic example is too artificial and simplistic for the experiments. Running the experiments on mere two real-world sequences and a single video-predictive model is rather minimal. After all, the approach is model-agnostic – why not experiment with multiple models?

Since STEP/EP-3D is closely related, I would expect reproducing the experiments in the original work and running the proposed approach against STEP/EP-3D on those benchmarks.

The readability and the technical exposition could be improved. For example:
* The definition in Eq. (2) contains a self-loop.
* STEP is first mentioned in l. 103 without any reference.
* Fig. 1 feels redundant. I’m confident all readers will be familiar with temporal continuity.


I encourage the authors to update the red-green colourmap in Fig. 2 and 5 to improve accessibility (congenital red-green colour blindness is quite common).

**Questions:**

* What video task do we assume in this work (action classification, video segmentation, etc.)?
* What would be an ideal explanation in the presented videos?
* How does the approach compare to STEP/EP-3D on the benchmarks presented in their original work?

---

### Official Review · Reviewer_X9Md · 2024-11-03

**Soundness:** 2
**Presentation:** 2
**Contribution:** 2
**Rating:** 3
**Confidence:** 3

**Summary:**

This paper proposes a novel video interpretation method based on perturbation. It considers the temporal dimension separately through dual perturbation, and finds a spatio-temporal visual explanation by conducting the TIS-aware spatial analysis for each frame based on extremal perturbation.

**Strengths:**

- This method improves the interpretation effect through the dual strategy of temporal and spatial perturbations, and performs particularly well in identifying key frames in dynamic videos.

- The experimental results show the effectiveness of TIEM in video interpretation.

**Weaknesses:**

- Although TIEM has made improvements in explaining the temporal dynamics of video models, its core idea is based on existing extreme perturbation methods and does not show fundamental innovation in methodology. This hardly meets ICLR’s high standards for methodological innovation.

- TIEM's experiments mainly focus on action recognition (UCF101-24 dataset) and artificially synthesized white-box models. Although the results are significant on these tasks, the generalization ability and applicability to a wide range of real-world application scenarios (such as medical video analysis, autonomous driving videos, etc.) have not been deeply explored.

- The baseline methods compared in the paper are not novel enough and lack comparison with the latest methods, such as AOSA in related work and other recently published methods.

**Questions:**

Please address the questions in the Weaknesses.

---

### Official Review · Reviewer_7DWU · 2024-11-03

**Soundness:** 2
**Presentation:** 1
**Contribution:** 2
**Rating:** 3
**Confidence:** 4

**Summary:**

This paper presents an approach to improve the explainability of video prediction models. In particular, the authors introduce two metrics, namely temporal importance scores (TIS) and TIS-aware importance map, to measure the importance of certain regions within certain frames with respect to the given video prediction task. Experiments on a synthetic video dataset and a real-world action classification dataset are conducted to verify the effectiveness of the proposed method.

**Strengths:**

- The problem of improving the interpretability of deep neural networks for video prediction is important.

&nbsp;

- Although some technical details are missing, the proposed method is straightforward.

**Weaknesses:**

- The proposed method seems largely ad-hoc.

  1) In Eq. (3), it is unclear what variable the maximization is over in the notation $\max(\theta)$. My understanding is that rather than enumerating all possible window sizes w, the authors use Eq. (3) to determine the proper value of w. However, $\theta$ is a function of w. If you maximize over w, I do not know what the computational benefit would be. More specifically, can you explain the computational complexity of the computation of TIS score?

  2) In Eq. (6), what is the definition of the lowest baseline bound \phi_0? It is introduced without proper explanation.

  3) In Eq. (7), what is the "vecsort” operator? Is it differentiable? If not, how are you going to minimize the loss?

&nbsp;

- The evaluation of the methods for the action classification task with real-world video datasets is not convincing enough.
  1) It is quite subjective to define the so-called signature frames. Does the dataset provide the signature frames of individual videos, or do you collect them? What are the statistics of such signature frames, e.g., spatial/temporal proportion over all frames? Given a new video prediction task, how do we determine them without or with less human annotation?

  2) The proposed “Temporal Pointing Game” metric does not consider the spatial extent within the signature frames. Even if there is only one pixel unmasked within the signature frames, the metric would be perfect.

&nbsp;

- The writing of the paper is quite confusing.

  1) In the introduction section, the description of the motivating example is confusing. For example, what is exactly the prediction task? Without a proper understanding of the problem setup, it is impossible to understand the meaning of the experimental results. Unfortunately, such detailed descriptions are delayed to the experiment section.

  2) The so-called STEP method is mentioned in the introduction section to motivate the problem. However, it was not introduced until the related work section.

  3) Although interpretability has been mentioned many times, the paper only considers the important spatiotemporal regions as a way of interpretation. I am not sure how much this would add up to the interpretability of the black-box deep models. In other words, what is the exact expectation for improving interpretability? Are the proposed quantified measurements really what people care about when they use video prediction models?

**Questions:**

I wrote the questions within the weakness part.

---

### Official Review · Reviewer_LRcA · 2024-11-04

**Soundness:** 1
**Presentation:** 1
**Contribution:** 1
**Rating:** 3
**Confidence:** 4

**Summary:**

Deep networks can classify actions in videos or predict future frames with high accuracy but remain very difficult to interpret.
A model agnostic approach is proposed that highlights which part of an input image-sequence is most important to a network's prediction.
Specifically, given a trained network and a test video, this perturbation based procedure generates a spatio-temporal mask which highlights what part of the input most influence the networks prediction.

**Strengths:**

This work tackles a challenging and important research direction: explaining the performance of trained deep networks.
The focus on video processing is timely and visual explanations have the potential to help understand networks better.
The paper has a clear progression, starting with a simple toy example and then describing a case study on real videos.

**Weaknesses:**

The paper's logic is weak because it rests on many unusual and loosely defined concepts: "temporal concentration", "temporal spillover", "gentle videos", "white-box regressor", "pointing game", etc. It would help to make these definitions more precise and to anchor your work in the well developed motion estimation literature in computer vision and signal processing (e.g. motion energy).

The stated aim of the paper is to improve the interpretability of video processing networks by generating visual explanation. But the method is quite complex and as a result the generated visual explanations are unintuitive. In particular the procedure described here involves many choices that are not clearly justified (e.g. why separate space and time, how are the many hyperparameters set, etc.).

Overall, the writing is unclear and the difficult to follow (e.g. description of the methods in Section 3). The paper contains many non-standard acronyms, some of them used before they are defined (e.g. STEP on line 103, only defined line 163). Some sentences do not make any clear point, for example line 222-224: "This separate learning structure of TIEM allows it to overcome the challenges of video interpretation by avoiding over-integration of the spatial and temporal dimensions during the interpretation and by focusing on temporal dynamics more explicitly."

The experiments are very limited and it is unclear what can be learned from the results. Can you show an example application of your method where we learn something unexpected that is not apparent using other visualization methods? What would your method reveal if you applied it to different networks and compared the visual explanations?

**Questions:**

The examples visual explanations shown in Fig 5 and Fig 6 all have a characteristic feature: they look like the combination several disks (or Gaussian distributions) at different spatial locations. Is there a constraint on the shape of the regions that your method can find? This was not discussed in the paper but seems to be implied by the figure and I do not know what to make of it.

In table 2, 3 and 6, you give a quantitative performance comparison on four example videos for which you have some ground truth to compare to. The "pointing game" metric is unintuitive, and you might want to consider calculating something more well known, like an f-score for example. Moreover, I do not understand where the error bar come from. You describe a deterministic procedure applied to deterministic networks. Where is the variability coming from?

You call your method perturbation-based, but Eq. 2 looks like a coarse approximation of a differential. Have you considered calculating the input-output jacobian of the network? Visualizing its dominant eigenvector would provide easily interpretable visualization of what the network is sensitive to.

---

### Official Review · Reviewer_aCrK · 2024-11-04

**Soundness:** 2
**Presentation:** 3
**Contribution:** 2
**Rating:** 3
**Confidence:** 4

**Summary:**

This paper proposes a perturbation-based method to evaluate the importance of spatial regions and temporal frames in a video.

**Strengths:**

1. The writing is generally clear, with easy-to-understand figures.
2. The idea is simple and straightforward to understand.

**Weaknesses:**

1. **Runtime of Perturbation**: What is the runtime of the perturbation process? It seems that numerous for-loops are required to exhaustively obtain importance scores for all configurations (e.g., different window sizes). Is this fast?

2. **Justification of Zeroing-out**: I understand that zeroing out certain frames or spatial regions is a seemingly natural way to create masked video data. However, could you provide additional justifications or insights for this choice? Why choose all zeros as opposed to other pixel intensity values? Importantly, why does this design become "model-agnostic"?

3. **Specific Video Classification Problem**: Why do we suddenly consider a more specific video classification problem after introducing Equation (2)? Isn’t the method supposed to work generally for any video prediction tasks?

4. **Computation of $\theta^w$**: How is $\theta^w$ computed exactly? The differential operator is not accurate because $p$ is not differentiable with respect to $t'$.

5. **Toy Problem in Section 4.1**: The example in Section 4.1 is a toy problem and is not convincing. Reduce its content and include more experiments on impactful applications and datasets.

6. **Role of "Signature Frames"**: "Signature frames" play a crucial role in the experiments on real data. It seems this label was provided in the original dataset; how was this obtained?

7. **Insufficient Data in Table 3**: Table 3 includes results from only two videos, which is not convincing.

**Writing Suggestions**:

- Explain what **STEP** is in Figure 2 before mentioning it for clarity. Figures are too small, and there is no clear caption to aid understanding.
- Consider moving some of the introductory text from Section 3.1 to the beginning to clarify the underlying tasks more specifically.
- In Equation (2), the probability on the left-hand side should also have a subscript $c$ to indicate it’s for a specific class. Additionally, it’s better not to reuse the notation $p$ on both the left and right sides.
- The constraint on $m_t$ for each $t$, located below Equation (5), involves the total frames $T$; is this a typo?

**Questions:**

See above.

---

### Note · Authors · 2024-11-26

I have read and agree with the venue's withdrawal policy on behalf of myself and my co-authors.